# A geospatial analysis of accessibility and availability to implement the primary healthcare roadmap in Ethiopia

Fleur Hierink [1,2✉], Olusola Oladeji[3], Ann Robins[4], Maria F. Muñiz[5], Yejimmawerk Ayalew[4] & Nicolas Ray [1,2]

## Abstract

**Background** Primary healthcare (PHC) is a crucial strategy for achieving universal health coverage. Ethiopia is working to improve its primary healthcare system through the Optimization of Health Extension Program (OHEP), which aims to increase accessibility, availability and performance of health professionals and services. Measuring current accessibility of healthcare facilities and workforce availability is essential for the success of the OHEP and achieving universal health coverage in the country.

**Methods** In this study we use an innovative mixed geospatial approach to assess the accessibility and availability of health professionals and services to provide evidence-based recommendations for the implementation of the OHEP. We examined travel times to health facilities, referral times between health posts and health centers, geographical coverage, and the availability and density of health workers relative to the population.

**Results** Our findings show that the accessibility and availability of health services in Somali region of Ethiopia is generally low, with 65% of the population being unable to reach a health center or a health post within 1 h walking and referral times exceeding 4 h walking on average. The density of the health workforce is low across Somali region, with no health center being adequately staffed as per national guidelines.

**Conclusions** Improving accessibility and addressing healthcare worker scarcity are challenges for implementing the primary care roadmap in Ethiopia. Upgrading health posts and centers, providing comprehensive services, and training healthcare workers are crucial. Effective outreach strategies are also needed to bridge the gap and improve accessibility and availability.

## Plain language summary

Access to primary healthcare, which encompasses essential healthcare services and often the initial point of contact between individuals and the healthcare system, is crucial for addressing the health needs of a population. In Ethiopia, ongoing efforts to reform the primary healthcare system aim to increase geographic access to health services and improve the availability of healthcare workers. This study focuses on the Somali region of Ethiopia and finds that 65% of the population is unable to reach a health center within 1 h of walking, and none of the health centers meet national and international staffing guidelines. These results play an important role in identifying areas where mobile outreach, involving trained service providers traveling to communities with limited access to healthcare facilities, can bridge gaps in healthcare accessibility and availability. Furthermore, the findings inform the implementation of primary healthcare reforms.

[1] GeoHealth group, Institute of Global Health, University of Geneva, Geneva, Switzerland. [2] Institute for Environmental Sciences, University of Geneva, Geneva, Switzerland. [3] UNICEF Ethiopia, Jijiga Field Office, Jijiga, Ethiopia. [4] UNICEF Ethiopia, Country Office, Addis Abeba, Ethiopia. [5] UNICEF, Eastern and Southern Africa Regional Office, Nairobi, Kenya. ✉email: fleur.hierink@unige.ch

Ensuring access to primary healthcare is an essential pre-requisite for achieving Universal Health Coverage (UHC), one of the key targets of the United Nations Sustainable Development Goals (SDGs) for 2030[1,2]. Globally, large gaps and spatial inequalities persist in accessing primary healthcare[3–7], with many communities in sub-Saharan Africa facing considerable distances from healthcare services, especially in sparsely populated rural areas[8]. The burden of inadequate and low-quality healthcare disproportionately affects low- and middle-income countries (LMICs), leading to preventable deaths[9,10]. Estimates suggest that in 2016, 8.6 million deaths in LMICs could have been avoided if accessible and quality healthcare services were available[10]. Being one of the least-developed countries on a global scale and the second most populous country in sub-Saharan Africa, Ethiopia faces notable challenges in healthcare delivery[11,12]. Despite notable improvements in Ethiopia's health indicators, the country still grapples with challenges such as a heavy burden of communicable and non-communicable diseases, persistent malnutrition worsened by conflicts and natural disasters, insufficient healthcare coverage and quality, concerns about sustainable healthcare financing, the need for workforce enhancements, gaps in disease surveillance and monitoring, and inadequate preparedness for future health emergencies[11,12].

To overcome these challenges the Ethiopian government has been committed to reach UHC by strengthening the primary health care system[13,14] (Supplementary Fig. 1). In 2003, the Health Extension Program (HEP) was introduced in response to an evaluation of the country's Health Sector Transformation Plan I which determined that access to health care for most Ethiopians was limited[15]. While UHC has not yet been fully attained, the HEP has contributed toward progress in maternal and child health outcomes, communicable diseases, hygiene and sanitation, and health care seeking[14,16,17]. The HEP's main goal was to expand coverage of the Ethiopian health system through the provisioning of health services by health extension workers (HEWs) from health posts (HPs), particularly to reach remote communities[15]. Health posts serve as an operational center for HEWs, and 5–8 HPs together with a health center (HC) function as a primary health care unit[14]. In Ethiopia, HCs provide curative and preventative care and are referral and logistic facilities for HPs. Health posts are the lowest level health facility in the Ethiopian health care system and provide mostly preventive care such as vaccination, family planning, and maternal health services[18]. Health extension workers are an expansion of the HPs operations and among other things manage home visits, outreach services, and refer patients to HCs[14]. The implementation of the HEP has led to an important increase in human resources for health and expanded access to health services in the country. The program has grown to include 18 packages of essential health services tailored to various contexts, such as pastoralist, urban, and rural areas. As a result, the number of HEWs and HPs has risen dramatically, from 2737 and 4211 in 2004 to 39,878 HEWs and 17,587 HPs in 2019[15]. In addition, the density of health personnel has improved notably, going from 0.3 physicians and 2 nurses per 10,000 population in 2005 to 11 health workers per 10,000 population in 2016, which is among the highest in Africa[19].

Due to challenges related to the maintenance of the HEW program and the scale-up of the primary health care system, the Ministry of Health designed a new transformative roadmap for the further development of the HEP from 2020 to 2035[15]. This roadmap, referred to as Optimization of Health Extension Program (OHEP), aims to better address the current epidemiological shifts in the country and to expand essential services to remote communities. This further expansion of the primary health care system requires new cadres to improve the availability, distribution and performance of health professionals to achieve UHC[15].

An important aspect of the roadmap is to restructure health service delivery by mapping HPs in relation to HCs, converting farthest HPs into comprehensive ones that can provide some essential HC services, and merging HPs that are too close to an HC as a unit into the facility. In pastoralist areas, such as those found in Somali Region, HEWs and mobile health services are used to fill gaps in access to health care. To ensure that the expansion of primary care does not compromise the quality of care[17], the current availability of health workers needs to be assessed at facility level to identify areas of understaffing. All health system transformations, as outlined in the OHEP, require an extensive geographical mapping and modelling to assess the current geographic accessibility of health facilities.

Somali region in Ethiopia is among the regions with lowest healthcare access[9]. In Somali region the population is mainly pastoralist (85%) and lives in remote and hard to reach areas that remain largely uncovered by the health system. The region has relatively large health disparities, with lower vaccination coverage, lower antenatal care coverage, higher infectious disease burden, and lower skilled birth attendance as compared to the national average[20–23]. Compounding these challenges are the region's weak infrastructure, a scarcity of health facilities, a shortage of qualified healthcare professionals, and an uneven distribution of the workforce, which collectively pose substantial hurdles to healthcare delivery[13,20]. Consequently, the residents of this region are particularly susceptible to adverse health outcomes, which led the Ethiopian government to classify this region as one of the four Developing Regional States (DRS)[23]. This heightened vulnerability underscored the importance of studying this particular region for the research.

Previous research on geographic accessibility to health services in sub-Saharan Africa[4–6,24–30], including studies conducted in Ethiopia[9,13,20], has shed light on the concerning patterns of inadequate spatial accessibility. However, these studies have either focused on other countries[25–27,29,30], primarily focused on continental trends and overlooked sub-national variations in health-seeking behavior[4–6,24], which is particularly relevant in pastoralist settings. Furthermore, these studies often failed to consider accessibility to primary health care services, and rather focused on emergency or obstetric care[5,24], or were conducted at a coarse resolution[6]. In addition, most of the research did not incorporate an analysis of the health workforce density and availability in conjunction with geospatial analyses. While recent scientific efforts have assessed accessibility and staffing sufficiency in Ethiopia[9], they lack precision at the facility level (e.g., current staffing) and are aggregated at larger sub-national units, especially for Somali region, making them unfit for the implementation of the OHEP roadmap.

To strengthen primary health care and reach of services in Somali region, where the population is mostly pastoralist and there are frequent public health emergencies, the United Nations Children's Fund (UNICEF) has supported the regional health bureau (RHB) for the past 14 years in deploying mobile outreach linked to routine health services[20]. UNICEF has also been supporting the operationalization of the OHEP Roadmap[15]. Our study aimed to provide evidence-based recommendations for the restructuring of service delivery platforms for the HEP in the Somali region. To this end, we developed a framework that considered multiple factors influencing health service accessibility and workforce availability. Specifically, we examined referral times, geographical coverage of health facilities, and the availability and density of health workers relative to the population. Our findings, which identified areas with the greatest gaps in accessibility and availability, have been used to inform the

restructuring of the HEP service delivery platforms and service packages under the new OHEP. This is one of the first studies to examine these factors in Somali region, Ethiopia, and could serve as a model for similar efforts at the national level and in other countries.

## Methods
Geographical accessibility was measured in terms of (1) travel time to the nearest health facility, (2) referral travel time between HPs and HCs, and (3) the proportion of the population covered by individual health facilities. In our analyses, we considered both motorized and walking speeds, along with a separate scenario dedicated to walking. However, when addressing referrals, we focused exclusively on the walking scenario. The reasoning behind this choice is elaborated further in the subsequent section. We used AccessMod[31] to create a travel impedance surface and calculated accumulated travel time and referral times using the least-cost path algorithm. We combined travel time data with population distribution data to extract population coverage statistics. However, merely analyzing spatial access to healthcare does not capture the full concept of access. Therefore, we combined our spatial accessibility models with data on available human resources at each HC in the region.

**Study area**. Accessibility to healthcare was assessed for Somali region, a region in the Eastern part of Ethiopia. This region is overall sparsely populated with 6 million people (Central Statistical Agency, 2013) spread over ~350,000 square kilometers, the second largest region in Ethiopia in terms of land mass, and predominantly nomadic pastoralists (85%)[20]. There are 99 woredas, or districts (second level administrative division) in this region. All 99 were included in the study.

**Data sources and preparation**. To create the travel impedance surface in AccessMod, we combined spatial data on digital elevation, land cover types, roads, rivers, and health facility coordinates. The digital elevation model was obtained from the Shuttle Radar Topography Mission (SRTM)[32], and the land cover raster was sourced from Sentinel-1 (2016)[33]. Data on roads were extracted from OpenStreetMap (2019)[34] and enriched with information on road conditions from the Ethiopian Federal Road Authority (2008). Hydrography, including line- and polygon shapefiles of rivers and lakes, were drawn from OpenStreetMap (2019)[34]. The population distribution data was taken from WorldPop[35] for the constrained United Nations adjusted population estimate for 2020. All data were projected in the same spatial reference system (UTM 38N, EPSG:32638) and processed in Quantum GIS (v3.16) and R (v4.1.1). The landcover and Digital Elevation Model (DEM) raster were resampled and aligned to the spatial resolution and extent of the population count raster (~92 meters). All separate spatial datasets were uploaded to AccessMod and combined into one merged landcover. Rivers and lakes were treated as full barriers to the population unless a road was crossing. We moved facilities slightly outside the country boundaries to the nearest grid cell within the country, and moved facilities located in water bodies to the nearest grid cell on land. The presence of health facility locations situated on landscape features that pose barriers to the population, such as rivers or lakes, can be attributed to two factors. Firstly, the terrain features used in our analyses, including rivers and roads, are rasterized at an approximate resolution of 100 meters. In certain cases, this resolution may slightly overestimate the width of rivers, leading to the placement of facility coordinates on barriers. Secondly, there may be slight imprecisions or inaccuracies in the GPS coordinates of the facilities. To mitigate the potential impact of these factors on our analyses, we adjusted the facility coordinates by relocating them to the nearest land grid cell.

We developed travel scenarios with inputs from the UNICEF zonal office in Jijiga (Supplementary Tables 1–2). Two different scenarios were considered, one where people walk to the nearest road and transfer to a readily available motorized vehicle (motorized + walking), and one where they only walk. We calculated travel time to HCs and HPs together and to HCs only. Referral times between HCs and HPs were modeled using a walking-only scenario, as it was assumed that most of the population does not have access to motorized vehicles and lives in pastoralist communities. Referral travel times were calculated in the direction of HPs to HCs and always took into account the nearest health facility (even beyond the woreda borders). The primary mode of transportation for referrals from HPs to HCs was walking, as transportation options are scarce in most communities. Ambulance services are exclusively available for referrals from health centers to hospitals. The geographic coverage tool in AccessMod was used to model the catchment areas of HCs and HPs based on 1 h and 2 h walking travel times. One-hour and 2 h catchment areas hold importance due to the recognition by clinicians of the "golden hour" during which immediate medical care is crucial in preventing death[36], as well as the 2 h threshold for emergency obstetric care[37,38]. The full catchment area of each facility was also modelled, this was determined using the cost allocation option in AccessMod, which calculates the area nearest to a specific facility in terms of travel time. This approach is similar to the way Voronoi diagrams are delineated using Euclidian distance methods[39]. However, this paper primarily focused on catchment areas with travel times of 1 and 2 h.

**Health facility and human resource data**. We obtained data on the geographical location of public health facilities (including health posts and health centers) and human resources from UNICEF and RHB in Somali Region, Ethiopia. The data were originally collected in 2020 and included information on various health worker groups at HCs, and only the number of HEWs at HPs. Health center staff include general practitioners, nurses, midwives, health officers, pharmacists, health information technicians, and laboratory staff. We modeled the number of people within a 1 h and 2 h travel time catchment area for each facility, allowing us to calculate the number and proportion of people with access to specific health worker groups and identify areas of low coverage.

The WHO has modeled the staffing requirements per group of health workers to achieve a UHC service index of 70%[40]. This index serves as a proxy for assessing the extent of health service coverage of essential health services (e.g., maternal & newborn care, infectious diseases, and non-communicable diseases) and is reported on a scale from 0 to 100. As an example, an increase of one unit in the coverage index is associated with an average need to increase the Health Worker Force (HWF) density by 9.2%[40]. We compared these WHO benchmarks with the density of health workers in the catchment area of 1 and 2 h around health facilities, assuming that if access to a health professional is needed, the service should be available and accessible in a timely manner. The national Ethiopian and the African WHO staffing requirements per health worker group are outlined in Supplementary Tables 3–4. Since the classification of health worker groups for the WHO staffing benchmarks differs from the Ethiopian classification, we reclassified the health worker groups according to Supplementary Table 5. For the calculation of staffing sufficiency as per the Ethiopian guidelines set out in the

national and regional Human Resource for Health Strategic Plan 2016–2025, we kept the classification as found in the data. The national and regional benchmarks for the health workforce indicate that each HC needs to have a minimum number of health workers to serve a population of 15,000–25,000 people (Supplementary Fig. 1 and Supplementary Tables 3–4). We also modeled staffing sufficiency per health center according to the Ethiopian government guidelines (Supplementary Table 3).

**Data charting and analysis**. The outcomes of the accessibility models, including the travel time rasters, the referral times, and the geographical coverage of HCs and HPs were visually mapped using QGIS (v3.16). An automatized workflow was created using the QGIS Atlas feature. Using this feature enabled the creation of individualized reports for all 99 woredas which were printed and shared with all facilities in Somali region to allow planning of outreach services and inform restructuring of facilities according to new OHEP guidelines. The descriptive and analytical quantitative data charting was done in R (v4.1.1)[41].

**Reporting summary**. Further information on research design is available in the Nature Portfolio Reporting Summary linked to this article.

## Results

**Accessibility of health facilities in Somali region**. Overall accessibility in Somali region is low (Fig. 1a–f), with 65% of the population unable to reach a HC or a HP within 1 h travel time (walking-only scenario). When considering only HCs, 86% of the population is unable to reach one within 1-h travel time (walking-only scenario). While accessibility improves slightly when considering motorized transport on roads, 56% of the population remains uncovered by HCs within 1 h of travel and 41% of the population is not covered by either a HP or HC within 1 h of travel by motorized transport. The percentage of the population able to reach a HC or HP in 1 h walking, ranges from 0% to 81.2% across woredas. The woreda with the lowest coverage is Wangey, where none of the population can access a HC or a HP in 1 h travel time and only 4.9% in 2 h travel time (walking-only). Given the high number of referral links between HPs and HCs (n = 1049), we decide not to visually map. Referral are calculated considering both physical distance (in kilometers) and time. The inclusion of the distance metric is essential as it aligns with the standard measure for implementing the OHEP roadmap. Referral travel times between HPs and HCs are on average 4 h and 45 min walking. However, the highest referral time, from Seley HP to Guradamole HC, is 28 h and 30 min (walking-only). The average referral distance is 17.9 kilometers in Guradamole, ranging from 0 to 107 kilometers.

Of all referrals, 70% of the HPs are located further than 10 kilometers away from the nearest HC, which according to the OHEP roadmap are candidates for upgrading into comprehensive HPs offering a wider package of services. Only 13 and 24 out of 1049 HPs are within one and two kilometers, respectively, of the nearest HC, suggesting that these could potentially be merged with the nearest health center. On the representativeness of travel distance for travel time, even if a HP is closer than 10 kilometers to the nearest HC, the travel time can take up to 3 h and 10 min. All referrals between 5 and 6 kilometers take at least 1 h and 15 min walking to be completed and maximum around 1 h and 50 min.

Health centers, providing slightly more specialized care than health posts, are highly inaccessible to the population when considering health seeking by foot (Supplementary Fig. 2A). Only 35.5% of the Somali population are able to reach a health center

in 3 h travel time, sharply contrasting the 80.4% who are considered to reach a health center in 3 h by motorized transport on roads (Supplementary Fig. 2B). Differences between the two scenarios are evident and show a sharp reduction in geographical access for both HCs and HPs when considering health seeking by foot, which is assumed to be the primary mode of transport in Somali region.

The Ethiopian government has goals and objectives to improve access to primary healthcare, including benchmarks for facility coverage and the density of the health workforce. The Ethiopian Health Sector Transformation Plan for 2016–2020 outlines that a health center should cover between 15,000 and 25,000 people in rural settings, as in Somali region (Supplementary Fig. 1). Although these targets do not consider travel time, we find that the coverage of HCs in 1 h and 2 h walking catchments is much lower, with the majority of HCs, 34% in 1 h and 32.2% in 2 h, covering between 2000 and 5000 people (Fig. 2). Only 9% and 16.5% of the HCs cover more than 15,000 people in 1 h and 2 h walking travel time respectively (Fig. 2). In Jijiga, Shinile and Godey, four HCs serve a population ranging from 47,087 to 284,334 within 1–2 h walking distances. The high population density in these facility catchments in urban areas may impact the availability of health services.

While the majority of HCs have a much smaller catchment population than envisaged in the Ethiopian Health Sector Transformation Plan for 2016–2020 (Fig. 2) and could in practice serve a larger population, a small number of HCs potentially operate well above their capacity to meet the health care demands of the catchment population within 1- and 2 h travel time limits. This is especially true when considering the full catchment (i.e., the catchment area not limited by a travel time threshold, see methods) of the health centers (Fig. 2). However, only a small percentage of the Somali population is located within 1 or 2 h of the full network of public health centers (14.0% and 22.5%, respectively). As a result, around 7.7 million and 7.0 million people do not have access to a health center within these travel time limits. Nevertheless, it's important to note that health posts, particularly those with health extension workers, are able to provide healthcare to 1.7 million and 3.3 million people, additional to the population already covered by HCs, within 1- and 2 h travel times, respectively. Health posts thus do cover an additional 18.3% and 37.0% of the Somali population. While it's important to note that health centers and health posts offer different services and work together in synergy, health posts are particularly important for extending access to healthcare in rural areas and areas that are far from health centers.

**Availability of the health workforce at health centers in Somali region**. Most global and regional health workforce benchmarks are built on static densities that present the number of health workers per 10,000 population and not on travel time-based metrics. In Somali region, we compared the health workforce densities in 1 h and 2 h catchments to the guidelines set out in the national and regional Human Resource for Health Strategic Plan 2016–2025 (Supplementary Table 3) and to the WHO health workforce benchmarks[1] (Supplementary Table 4, second column).

In Somali region there is not a single HC that is adequately staffed in all categories of health workers as per Ethiopian government guidelines. The vast majority of HCs (95.4%) are understaffed in at least 6 out of 11 health worker categories. The greatest staff shortages are among pharmacists and laboratory technicians with university degrees. The only categories of health workers in which most HCs are adequately or overstaffed are druggists and laboratory technicians with diplomas, although a

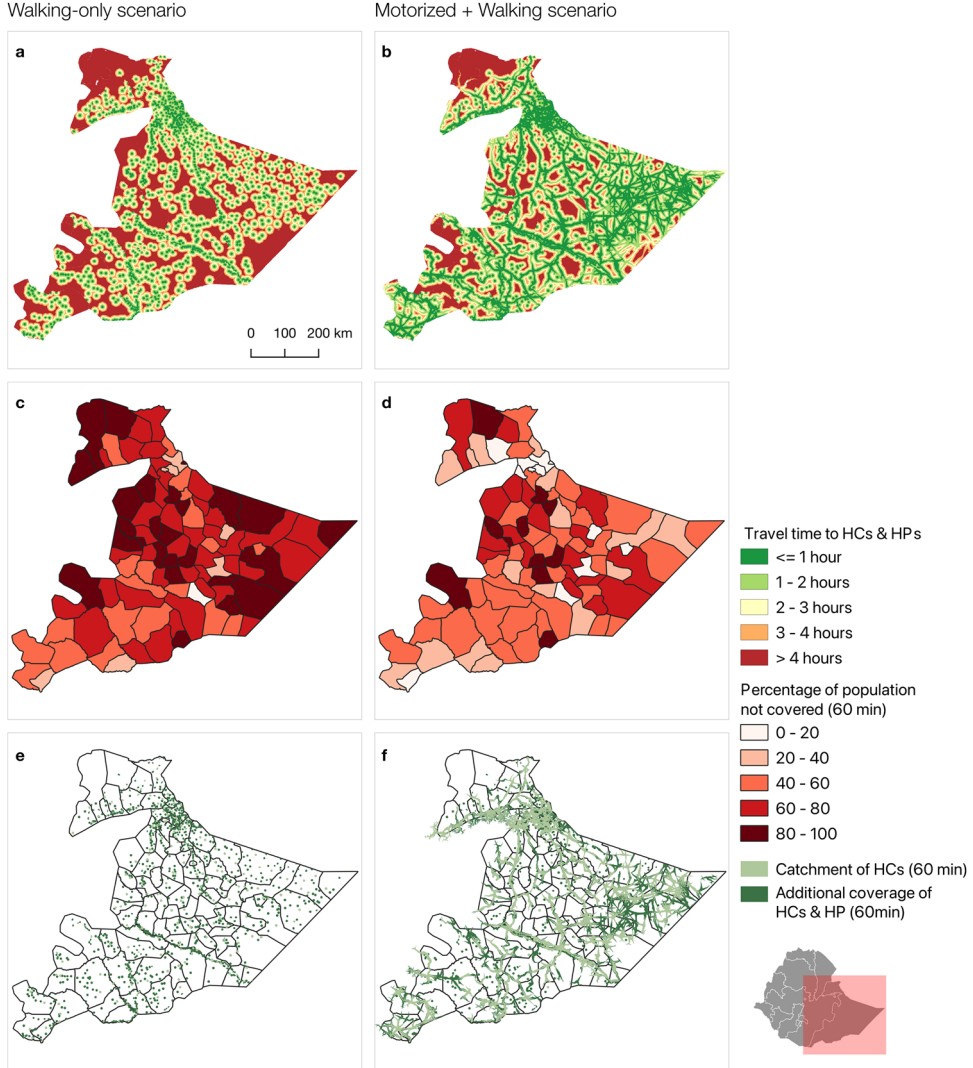

**Fig. 1 Accessibility to health facilities in Somali region.** Travel time to health posts and health centers considering (**a**) a walking-only scenario, (**b**) a motorized and walking scenario. Percentage of population remaining uncovered by health centers and health posts in 1-h travel time considering (**c**) a walking-only scenario, (**d**) a motorized and walking scenario. Geographical coverage of health posts and health centers in 1-h travel time considering (**e**) a walking-only scenario, (**f**) a motorized and walking scenario.

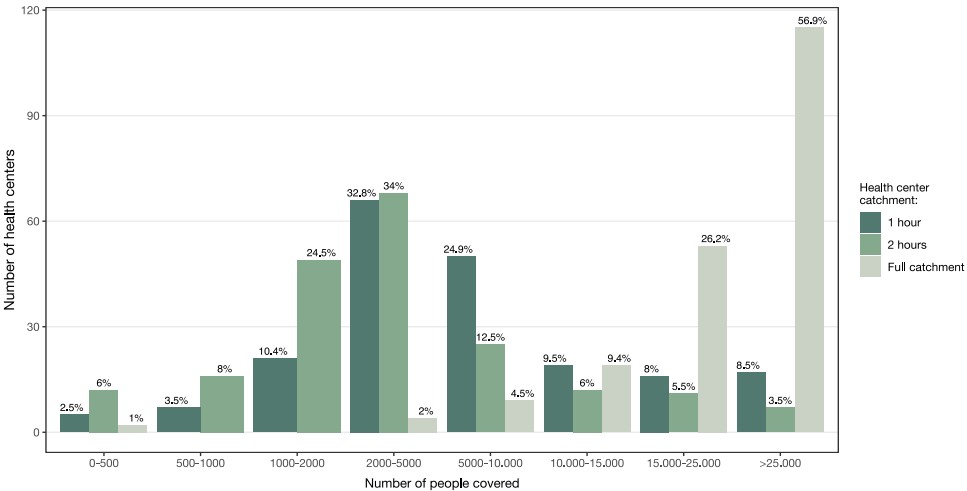

**Fig. 2 Coverage of health centers within 1- and 2 h walking catchments in Somali region.** Bars indicate the number of health centers per coverage category. Colors distinguish the 1 and 2 h walking catchments. Percentages reflect the proportion of health centers in a specific coverage class.

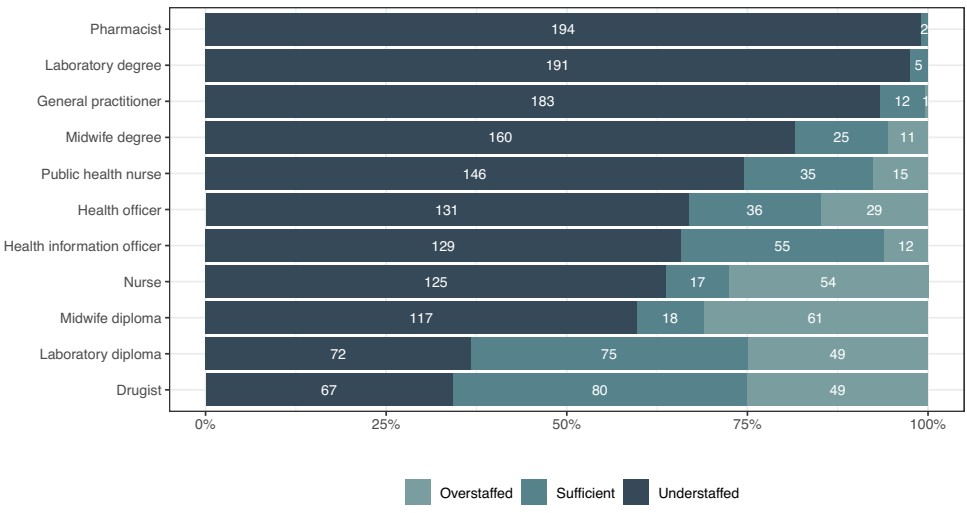

**Fig. 3 Staffing sufficiency for all health centers per health worker category.** Numbers represent number of health centers per category.

large number of HCs still remain understaffed in these categories (34.2% and 36.7% respectively) (Fig. 3). When looking at staffing sufficiency in relation to the size of the 1 h population catchment area, there are fewer HCs with small catchment areas that are sufficiently and overstaffed, while there seems to be a tendency for slightly more HCs with large catchment areas to be overstaffed. Understaffing, on the other hand, occurs in various degrees of 1 h catchments, but slightly less in health centers with larger catchments. The geographical distribution of the additional health workers needed in understaffed HCs does not follow any particular pattern. However, the number of health workers needed per health center seems to be generally lower in the northern (urban) Somali region (Supplementary Fig. 3).

When comparing the health worker densities in 1- and 2-h walking catchments to the WHO benchmarks, which in contrast to the Ethiopian Health Sector Transformation Plan for 2016–2020 are linked to population density, we find that for all health worker groups except for medical assistants (health officers), the availability is far below the required thresholds (Fig. 4a, b). There is especially a dire need for additional nursing and midwife professionals, where respectively 5640 and 10,111 health workers are needed in 1 h and 2 h catchments to reach the WHO benchmark (Fig. 4a, b). However, large investments in additional health workers are also needed in the other groups, where between 963 and 2771 health workers are required to reach the benchmark in 1 h and 2 h catchments. Even when the density of health workers is sufficient to serve the population within 1 h or 2 h walking catchments (i.e., 14% and 22.5% of the population), many people in Somali region live beyond these catchments and therefore have limited access to static health services provided at health centers. This emphasizes the importance of health posts, health extension workers, and mobile health and nutrition teams who can bring essential healthcare services to remote areas and increase accessibility for these communities. The figures presented here only refer to the catchment areas of the HCs and do not represent the effective range of services added by HPs and HEWs.

The need for additional health workers to meet the WHO benchmark varies spatially (Fig. 4b and Supplementary Fig. 4). Of all additional health workers required, proportionally and spatially the highest need for additional health workers is in Jijiga (indicated in Supplementary Fig. 4a), the capital city of Somali Region, where 19.4% of the total additional required nurses (Fig. 4b), 18.2% of the total additional required pharmacists, 18.1% of the total additional required laboratory

technicians, and 16.6% of the total additional required medical doctors are needed to cover the population falling in 1 h HC catchments (Supplementary Fig. 4). The spatial distribution of the required additional health officers classified as medical assistants varied widely, with densities higher than required in several woredas, implying that the distribution of the medical assistants could be optimized to ensure a larger coverage.

**Availability of health extension workers in health posts in Somali region.** We have information on the number of health extension workers for all health posts in Somali region. According to the Ethiopian government guidelines, two HEWs are required per health post. Only 29.6% ($N = 311$) of all HPs meet this requirement, with more than 50% of HPs having only 1 health extension worker. Comparing the woredas shows that Jarati requires the highest number of additional HEWs ($N = 49$). Figure 5 shows that even in woredas where a large proportion of the population is further than a 2 h walk from a health center or health post, there are large gaps in the availability of HEW for outreach activities. In two specific woredas, more than 75% of the population resides beyond a 2 h distance from both health posts and health centers. Consequently, the presence of HEWs is vital to ensure healthcare access for these remote communities through conducting outreach activities. However, it is precisely in these two woredas where there is a shortage of between 25-30 and 30-40 HEWs. Focusing only on woredas where less than 50% of the population can walk to a HC or HP within 2 h, there are 33 woredas that need between 1 and 33 (average 9) additional HEWs each. In total, 798 additional HEWs are needed across Somali region to meet government targets. This figure does not take into account the 11 woredas that have a total surplus of 64 HEWs (Fig. 5). In addition, the government guidelines do not reflect adjusted standards for woredas with low coverage, where a higher number of HEWs and outreach activities by mobile brigades may be needed to meet the health care needs of the population.

**Discussion**
The accessibility and availability of health services in Somali region of Ethiopia is generally low. The majority of the population lives far beyond the 1 h and 2 h travel time limits when seeking health care by foot. The geographical accessibility of health services tends to be higher in urban centers such as Jijiga, but the availability of health workers is lowest there due to the

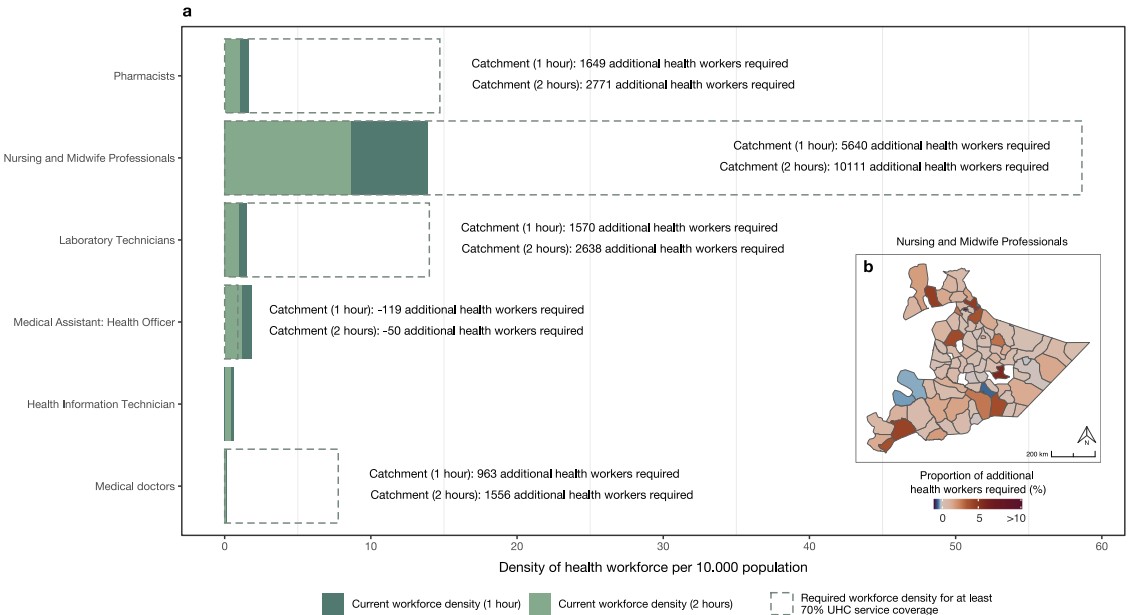

**Fig. 4 Health worker densities in health centers in 1 h and 2 h catchments compared against the WHO required workforce density benchmarks[40].**
**a** Dashed line indicates the required WHO workforce density per health worker group. Filled bars indicate the found densities in 1 h and 2 h catchments respectively. Text represents the additional number of health workers needed per health worker group and in the different travel time catchments to achieve the WHO required workforce density. **b** Map showing where the required number of nursing and midwife professionals to reach the WHO benchmark[1] is proportionally highest in 1 h catchments. Red colors indicate a proportionally higher need of additional health workers. Blue indicates that the density of health workers is higher than required under the WHO benchmark.

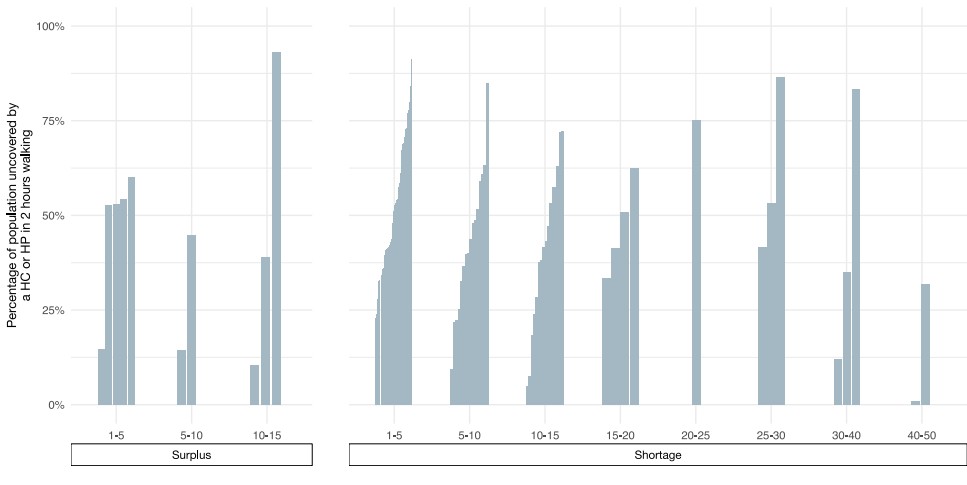

**Fig. 5 Availability of health workers (HEWs) per woreda.** Each bar shows a woreda where there was either a surplus of HEWs (left) or a shortage of HEWs (right). The height of the bar indicates the relative proportion of people uncovered by a HC or HP in 2 h walking. The x-axis shows either the number of surplus or missing HEWs.

higher population density and the low number of qualified health workers. The gap we found between geographical accessibility and availability demonstrates and underlines the importance of considering the different dimensions of access to health care rather than solely focusing on one. The scarcity of healthcare workers presents an important challenge for the successful implementation of Ethiopia's new primary care roadmap. It necessitates considering factors such as the proximity of health centers to health posts, addressing staffing capacity and gaps, and formulating sustainable and cost-effective outreach strategies to enhance both geographical accessibility and availability of human resources. Efforts should focus on bridging the gap between these two dimensions.

The main strength of this study lies in its, to the best of our knowledge, novel approach of coupling geographical accessibility models with facility-level information on health worker staffing. This integration allows for a more realistic assessment of geographical accessibility to specific health worker specialties and provides valuable insights into areas where the health workforce should be strengthened even at the facility level. In contrast, static measures such as the number of health workers per 100,000 population, which are often still the standard, fail to capture the actual geographic coverage of the health workforce. A recent study conducted by Hendrix et al.[9] also examined the conjunction of geographical accessibility with human resource availability in Ethiopia. However, in this study, the required human resource

availability for each facility type was considered to mirror the actual availability of healthcare workers. As demonstrated in this research, it is evident that the actual availability is frequently much lower. However, the results of Hendrix et al.[9] indeed also show high travel times and low access to healthcare for most of Somali region.

This study is among the first of its kind to compare the results against the benchmarks set by the WHO to achieve universal health coverage. The study has shown that even though geographic accessibility of the primary health care system may seem high, the actual availability of staffing to meet the demand in health care can be low. This is especially the case in urban areas where population density is high. This study highlights that solely focusing on geographical access fails to capture the field reality of people seeking care and has highlighted the importance of health extension workers that reach out to remote communities. Unfortunately, our study was limited by the lack of precise data on care utilization, which hindered our ability to quantify the population's specific service requirements and adjust the necessary additional health workforce accordingly. Combining geographic accessibility models with more detailed information on the availability of health services and health workers is key in realistically assessing gaps in service delivery.

**Policy implication of accessibility modelling in Somali region.** The main objective of this body of work was to inform the implementation of the OHEP roadmap and to identify where HCs and HPs are in too close proximity of each other and to know where facilities operate below staffing capacity[15]. Here, we discuss how this work can inform the evidence-based implementation of the HEP roadmap. One of the key reforms in the HEP is namely to stratify HPs into three categories of service delivery points: (1) comprehensive HPs (located beyond ten kilometers from the HC or primary hospital), (2) basic HPs (located within ten kilometers of an HC), and (3) HEP units in HCs (for villages with nearby HCs or primary hospitals)[15]. While literature and anecdotal field evidence suggested that potentially a high number of HPs is located too closely to HCs, our evidence shows that more than 70% of the HPs are located beyond ten kilometers of an HC and are to be upgraded into comprehensive HPs or would need additional mobile health and nutrition teams to reach pastoralist communities.

In Somali region, each of the 99 woreda has identified 3–4 HPs to be upgraded into comprehensive HPs and several HPs that are close to the HCs that will be closed and merged with the HCs in the first phase of the implementation of the OHEP roadmap, which is over the next 3–5 years. Costing including human resources for each woreda has already been completed in most woredas. For this study, we presented our results assuming that the population crosses woreda borders to seek healthcare, however for the OHEP implementation we conducted an additional scenario where we assumed for some priority woredas that health care is only accessed within the woredas. This is because the stratification of HPs into merged and comprehensive HPs is restricted to the distance between HPs and HCs within the same woreda. However, from the mapping presented in this study, some HPs are closer to HCs in neighboring woredas and are more suitable as referral health facilities. If the distance between a HP and the nearest HC is taken into account, irrespective of whether or not the HC is located in the same woreda as the HP, the number of HP that need to be upgraded to comprehensive HP can be reduced, saving costs, ensuring a more efficient and effective referral system, and strengthening collaboration between different woredas. This is particularly useful as our modeling shows that in Somali region more than 70% of the

HPs will need to be upgraded into a comprehensive HPs based on the criteria set out in the HEP which may not be readily achievable given logistics and funding availability. The mapping will help policy makers to assess the feasibility and practicability of the 10 km criterion in a pastoral setting like Somali region.

Another important reform in the HEP is the development of service standards for each type of HP. Communities living further than 10 km from a HP are to be provided with a comprehensive package of essential services. If the HP cannot be upgraded in terms of human resources and infrastructure; the same comprehensive package can be provided through mobile health services, although costing of this provision needs to be weighted. The results of this mapping can provide evidence-based priority areas for the deployment of mobile health teams, especially in woredas with a low coverage of health facilities.

**Study limitations & implementation challenges.** Somali region is predominantly rural[20] with a high proportion of pastoralist communities[42]. Nomadic pastoralists are among the most marginalized, inaccessible and underserved populations in the world, resulting in pastoralists being severely underrepresented in health planning data, including population censuses and large-scale health surveys such as the Demographic and Health Surveys (DHS) program[43,44]. The lack of accurate data on the pastoralist population, including their numbers and distribution, leads to potential uncertainty about their representation in gridded population datasets[44,45] and thus in our accessibility models. Our accessibility model relies on gridded population data from WorldPop[35] which constrains the distribution of the population to recognized buildings and settlements. However, these data are based on building and settlement traces obtained from satellite data[46,47]. As nomadic pastoralists are highly mobile and seasonal, they are not easily captured in settlement data and are unlikely to be well located and distributed in gridded population datasets. This implies that during some parts of the year the geographic coverage of health facilities in Somali region may fluctuate when either pastoralist communities temporally reside within the catchment of some facilities or fall far beyond the reach of the health system. Of course, these seasonal and temporal movements of pastoralist communities requires a coordinated plan for outreach to ensure access to essential health services[20,43]. Especially since the use of health services at static health facility structures is found to generally be low among pastoralist communities in Somali region[21].

We recognize that some of the catchment area assumptions we have made in this study are arbitrary and may not reflect the actual travel time limits when people seek health care. In a study conducted by Okwaraji et al.[18], the average travel time to reach maternal health services in a rural district in northwestern Ethiopia was found to be well over 2.5 h. This means that the threshold of 2 h travel time that we set in several parts of our study may in fact be much higher. However, in international guidelines, 2 h is often considered the normative upper limit to reach facilities providing emergency surgery and blood transfusion, notably in case of obstetrical complications[24,29]. To account for the potential uncertainty of the found health workforce densities in 1- and 2-h catchment areas, we used the cost-allocation output from AccessMod to delineate full non-overlapping catchment areas. The output of a cost-allocation analysis is a raster file indicating unique areas for each health facility to which a specific health facility is closest in terms of travel time. These areas were then considered to represent the full catchment area of a given health facility. We present the found full catchment population in Figure 2 and the health workforce density in Supplementary Fig. 5. Although these results provide

useful insights, they overlook one important factor, which is the outreach of mobile health units and health extension workers. In addition, our results are limited to health facilities classified as health posts or health centers, which make up the biggest part of the public primary healthcare system. Yet, there are approximately 15 governmental district hospitals and one regional referral hospital in Somali region, which are currently not included in our analysis because of lacking coordinates and human resource information. The health workforce densities presented here thus solely reflect the densities found in HCs and HPs. Expanding our main results to the full catchment areas may therefore present unrealistically low health workforce densities that in reality are higher because of the presence of facilities that go beyond the primary governmental facilities or are covered by outreach activities through HEWs and mobile health units. Furthermore, our human resource data for HPs was limited and focused solely on the number of HEWs, failing to provide a comprehensive reflection of other crucial health worker categories involved in service delivery at this level, such as nurses.

Capturing all factors related to health seeking, including travel mode, speed, cultural barriers, and proximity to higher levels of care like health centers (HCs) or hospitals, presents another notable challenge. Field evidence reveals that some communities might bypass the nearest HC or health post (HP) due to restricted movement during conflict or emergency situations. To ensure the accuracy and realism of accessibility models, particularly in pastoralist settings, it is crucial to understand these specific health-seeking behaviors and tailor the scenarios applied to the models accordingly. Moreover, while the inclusion of human resources in healthcare accessibility falls short of providing information on dimensions like affordability, acceptability, and accommodation, it does offer a more comprehensive understanding compared to solely considering geographical access.

## Conclusions

Practical outputs, such as maps and statistics derived from geographic accessibility models prove to be meaningful and useful decision-making tools to guide the implementation of the OHEP roadmap in Somali region, Ethiopia. The methods presented in this paper go far beyond the use-case of Somali region in Ethiopia and health care and can provide insightful information for the planning and distribution of schools and teachers for example[39]. In this paper, the geographical accessibility of health facilities is modelled in combination with the availability of human resources. Combining these two dimensions of health care access shows that in areas where geographic access of health facilities is relatively high, the availability of health workers can be poor, meaning that once a patient reaches a facility, they may not receive the care needed. Implementing the HEP guidelines and bridging the gap between accessibility and availability will require substantial funding and training, such as deploying additional health workers, planning outreach activities of mobile brigades, upgrading HPs to comprehensive HPs, and expanding HCs to include HP. The findings presented here help to progressively prioritize these investments by targeting woredas and facilities where the need is greatest.

## Data availability

The data used in this study include digital elevation model (DEM) from the Shuttle Radar Topography Mission (SRTM)[32], landcover data from Sentinel-1[33], road data from OpenStreetMap[34] and the Ethiopian Federal Road Authority, hydrography data from OpenStreetMap[34], population distribution data from WorldPop[35], as well as health facility and human resource data obtained from UNICEF and the Regional Health Bureau in Somali Region, Ethiopia. The datasets related to health facility location and staffing sufficiency are not publicly available due to the sensitivity and nature of the information but are available from the corresponding author on reasonable request. All other data sources are openly available and can be downloaded from the respective sources. Source data for the figures are available as Supplementary Data 1–5 and can also be downloaded from Zenodo (https://doi.org/10.5281/zenodo.8362737)[48].

## Code availability

The results presented in this paper are not based on custom code or mathematical models or algorithms. The code and data for the creation of Figs. 2–5 can be obtained from Zenodo (https://doi.org/10.5281/zenodo.8362737)[48].

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

## Acknowledgements
We would like to express our sincere gratitude to Samuel Abera for his invaluable assistance in providing us with a portion of the input data used in this study. His generous support and timely response to our request were crucial to the success of our research.

## Author contributions
F.H. led the study along with N.R. Conceptualization of the study was done by F.H., N.R. and O.O. Initial discussions on needed outputs and results were facilitated by M.F.M. A.R., Y.A. and O.O. Data on health services and staffing sufficiency were provided by A.R., O.O. and Y.A. The methodology was initially developed by F.H. and N.R. and reviewed by all co-authors. Data analysis and processing were done by F.H. and supported by N.R. Writing of the original draft was done by F.H. and supported by O.O. and N.R. Initial reviews on the outputs and results were given by O.O. and supported by A.R., Y.A. and M.F.M. The paper was edited by F.H., O.O., M.F.M., A.R., Y.A. and N.R. All authors have further assisted in thoroughly reviewing all figures and texts.

## Competing interests
The authors declare no competing interests.
