## [Peer Review File · Communications Medicine]

Reviewers' comments:

Reviewer #1 (Remarks to the Author):

Thanks for this opportunity to review this work. I have some suggestions as follows.

Introduction

Why did the authors choose Somali region in Ethiopia? Please provide brief justifications.

Before entering the detailed situations in Ethiopia, it is necessary to talk about background information happening in other countries. For example, see "Jia et al. Inequalities of spatial primary healthcare accessibility in China. *Social Science & Medicine* Volume 314, December 2022, 115458" for a similar, nationwide study. Please use references like this to place your study in the context. Also, compare your findings with those from studies from different developing contexts.

"We modeled the number of people within a one-hour and two-hour travel time catchment area for each facility" Please justify your selection of the catchment radii.

Are there any other existing (similar) efforts in other regions of Ethiopia? In other African countries?

What are the strengths and limitations of this study?

Reviewer #2 (Remarks to the Author):

This paper uses appropriate geospatial data and methods to identify health care shortages (and surpluses) in a rural region of Ethiopia. The main contributions are the methodology, which is applicable to other low-income countries where the uneven geographic distribution of health services is a significant challenge, and the demonstration of the importance of considering travel times to services in such contexts. Analyzing two travel modes – walking and walking & motorized – is also a strength. Despite these strengths, the paper often seems more like a policy report than a research publication. There are too many tables and figures, including supplementary ones, and the results section relies too much on information in supplementary material. This makes it difficult to discern the main scientific contributions. Detailed comments and suggestions are below:

The research is only weakly connected to the broader literature on geographic distribution of health services in low-income countries. In the introduction, clarify the contribution of the work to that literature.

The point on lines 220-221 about the global metrics used is very important. It would be good to mention that point in the introduction as one of the motivations for the work.

Explain the concept of referral time a bit more. Are people typically referred from an HP to a HC, and how frequently does this happen? Later you mention that referral travel typically occurs via walking, but I would have thought that transport of sicker patients, who (I suspect) would be referrals, would be more likely to occur via motorized if possible.

It would be good to include a workflow diagram for the methodology, so that the approach could be followed in other study areas where detailed geospatial data are lacking

Line 68 -- Mention whether travel times were based on walking or car/motorized. This is explained later, but briefly mention here also.

Lines 98-100 describe health facility locations in water, etc. Was this due to lack of precision in the location coordinates for some health facilities, or to the coarse spatial resolution of the land use data (grid cells). Explain briefly

The Results section is too detailed and relies too much on material in the many supplementary tables and figures. The section is overloaded with statistics that would be appropriate for a governmental report but are not very meaningful for a broad audience of researchers and policymakers. This is especially true in the workforce and workers sections. Tighten up this section by omitting some of the detailed statistics; eliminating many of the supplementary tables and figures; and focusing more on the overall patterns and trends.

In discussing referrals (starting on Line 166), you focus on distance, not travel time. Explain – was this done because distance is emphasized in the Roadmap? Also, explain whether distance is Euclidean or based on road/path network. The same paragraph then shifts to travel time, but it is unclear if travel time is based on walking or walking plus motorized.

Lines 238-243. Some of the workforce results are only presented in supplementary tables and only discussed very briefly. For readers, this can be confusing. Either omit some of the supplemental results, or explain briefly in the text. For example, the statement "... nurses make up the biggest group" is vague. Either delete or just give the overall statistic in the text – i.e. Nurses account for X% of the overall HC workforce. Figures 3 and 4 already contain a lot of good information, so just focus on those.

Lines 294-297 are difficult to follow and need to be rewritten. For example, the first sentence refers to the results for two woredas, and it is unclear what the value ranges (e.g. 25-30) refer to. The next sentence

The discussion section also focuses too heavily on detailed findings about specific health service needs in the study area and policy recommendations. Need to cut some of the details and focus more on general findings that may be relevant in similar kinds of places; lessons learned; strengths of the methodology; etc.

Minor edits:

Lines 42 and 43 – don't need to repeat the abbreviations (HP, HC) here, because they were provided on the previous page

Line 62 – change to: restructuring of the HEP

Lines 126-7 – Briefly explain the UHC Index – percent of what?

Fig. 1 e and f – It's hard to see the gray-tone shading. Might be good to switch to color shading.

Lines 213-15 – "additional" – compared to what? Compared to HCs only? Compared to the situation before HPs?

Lines 234-5 -- potentially to compensate ... unclear – rewrite.

Line 289 – change to: guidelines

Figure 5 – x-axis should be labelled: Number of HEWs (Surplus or Shortage)

Line 322 – optimization of??

Line 329 – change: from the TO of an

Reviewers' comments:

Reviewer #1 (Remarks to the Author):

Thanks for this opportunity to review this work. I have some suggestions as follows.

We thank the reviewer for the effort and time invested in reviewing our manuscript and for addressing some important suggestions and comments. We have made changes throughout the text accordingly. In the responses below we will indicate the changes made.

Introduction

Why did the authors choose Somali region in Ethiopia? Please provide brief justifications.

Response: *The Somali region in Ethiopia stands out as one of the most marginalized regions due to its pastoralist population, which leads to a significant portion of the communities residing in remote and inaccessible areas that lack adequate health system coverage. Furthermore, the region is classified by the government as one of the Developing Regional States, characterized by social indicators that are significantly below the national averages. The region is also vulnerable to various humanitarian crises, such as conflicts, droughts, floods, and disease outbreaks, including cholera and measles. These circumstances necessitate the presence of a robust and high-quality healthcare system capable of effectively responding to these challenges and providing essential healthcare services to the population in need¹. This marginalization prompted our study's specific focus on the Somali region. Moreover, the collaboration with UNICEF Ethiopia, a key partner involved in implementing mobile outreach and community health services in Somali, further solidified our interest in this region. By working together, we aimed to address the unique challenges faced by the population and contribute to supporting the implementation of the OHEP roadmap in the Somali region. We have changed the last part of the first paragraph to strengthen and justify this particular interest.*

"Compounding these challenges are the region's weak infrastructure, a scarcity of health facilities, a shortage of qualified healthcare professionals, and an uneven distribution of the workforce, which collectively pose substantial hurdles to healthcare delivery^{9,13}. Consequently, the residents of this region are particularly susceptible to adverse health outcomes, which led the Ethiopian government to classify this region as one of the four Developing Regional States (DRS)¹². This heightened vulnerability underscored the significance of studying this particular province for the research."

Before entering the detailed situations in Ethiopia, it is necessary to talk about background information happening in other countries. For example, see "Jia et al. Inequalities of spatial primary healthcare accessibility in China. Social Science & Medicine Volume 314, December 2022, 115458" for a similar, nationwide study. Please use references like this to place your study in the context. Also, compare your findings with those from studies from different developing contexts.

Response: *We have added the reference to Jia et al. (2022) and changed the sentence to read: "Globally, large gaps and spatial inequalities persist in accessing primary healthcare³⁻⁷, with many communities in sub-Saharan Africa facing considerable distances from healthcare services, especially in sparsely populated rural areas⁹." We feel that there are considerable contextual differences between China and Somali region in Ethiopia, making it challenging to compare the results. Yet we do agree that it is important to highlight that similar efforts have been done in other regions of the world, hence the addition of the reference to the introduction.*

"We modeled the number of people within a one-hour and two-hour travel time catchment area for each facility" Please justify your selection of the catchment radii.

Response: *We agree that no justification for these thresholds were given in the paper. We have therefore added the following sentence to the methods: "One-hour and two-hour catchment areas hold significance due to the recognition by clinicians of the "golden hour" during which immediate medical care is crucial in preventing death²⁶, as well as the two-hour threshold for emergency obstetric care^{27,28}."*

Are there any other existing (similar) efforts in other regions of Ethiopia? In other African countries?

Response: *We appreciate the reviewer's suggestion to explore the existing literature on health worker availability. While we are familiar with several recent studies on this topic in Ethiopia, such as those conducted by Hendrix et al. (2023), Woldemichael et al. (2019), and Oladeji et al. (2021), it is important to note that these studies primarily focus on health worker availability at aggregated administrative levels and do not specifically address the geospatial distribution of health workers at a fine scale and at the facility level. However, a study by Oliphant et al. (2021)*

conducted a geographic accessibility analysis to evaluate the accessibility of community health workers in Niger and identify areas where targeted optimization for the placement of new community health workers is required.

What are the strengths and limitations of this study?

Response: We sincerely appreciate the reviewer's inquiry. The main strength of this study lies in its novel approach of coupling geographical accessibility models with facility-level information on health worker staffing. This integration allows for a more realistic assessment of accessibility to specific health worker specialties and provides valuable insights into areas where the health workforce should be strengthened even at the facility level. In contrast, static measures such as the number of health workers per 100,000 population fail to capture the actual geographic coverage of the health workforce. Additionally, this study is the first of its kind to compare the results against the benchmarks set by the World Health Organization to achieve Universal Health Coverage. We have added an additional paragraph to the discussion to highlight the strengths accordingly.

Nevertheless, the study encountered several challenges. For instance, assumptions regarding the 1-hour and 2-hour catchment areas may not accurately reflect the health-seeking behavior for less severe health issues. However, we attempted to address this limitation by considering the calculation of the full catchment area, as discussed in the paper. Additionally, there are inherent data limitations, such as incomplete digitization of the road network, which may result in lower travel speeds in undermapped areas. We have acknowledged and discussed these limitations in the dedicated discussion section of the paper, aiming to sufficiently address the reviewer's question.

Reviewer #2 (Remarks to the Author):

This paper uses appropriate geospatial data and methods to identify health care shortages (and surpluses) in a rural region of Ethiopia. The main contributions are the methodology, which is applicable to other low-income countries where the uneven geographic distribution of health services is a significant challenge, and the demonstration of the importance of considering travel times to services in such contexts. Analyzing two travel modes – walking and walking & motorized – is also a strength. Despite these strengths, the paper often seems more like a policy report than a research publication. There are too many tables and figures, including supplementary ones, and the results section relies too much on information in supplementary material. This makes it difficult to discern the main scientific contributions. Detailed comments and suggestions are below:

We thank the reviewer for the thorough and positive feedback on our manuscript. We have carefully considered all the comments of the reviewer and made changes throughout the manuscript accordingly and in the point-by-point response below. We hope that the revisions are to the reviewer's satisfaction.

The research is only weakly connected to the broader literature on geographic distribution of health services in low-income countries. In the introduction, clarify the contribution of the work to that literature.

The point on lines 220-221 about the global metrics used is very important. It would be good to mention that point in the introduction as one of the motivations for the work.

Response: We have added an additional paragraph (see below) to the end of the introduction addressing both of the points mentioned by the reviewer and made sure that the information mentioned there is connected to the wider literature and also touches upon the point of the global metrics.

“Previous research on geographic accessibility to health services in sub-Saharan Africa^{4–6,19–25}, including studies conducted in Ethiopia^{9,12,26}, has shed light on the concerning patterns of inadequate spatial accessibility. However, these studies have either focused on other countries^{20–22,24,25}, primarily focused on continental trends and overlooked subnational variations in health-seeking behavior^{4–6,19}, which is particularly relevant in pastoralist settings. Furthermore, these studies often failed to consider accessibility to primary health care services, and rather focused on emergency or obstetric care^{5,19}, or were conducted at a coarse resolution⁶. Additionally, most of the research did not incorporate an analysis of the health workforce density and availability in conjunction with geospatial analyses. While recent scientific efforts have assessed accessibility and staffing sufficiency in Ethiopia²⁶, they lack precision at the facility level (e.g. current staffing) and are aggregated at larger sub-national units, especially for Somali region, making them unfit for the implementation of the OHEP roadmap.”

Explain the concept of referral time a bit more. Are people typically referred from an HP to a HC, and how frequently does this happen? Later you mention that referral travel typically occurs via walking, but I would have thought that transport of sicker patients, who (I suspect) would be referrals, would be more likely to occur via motorized if possible.

Response: People are referred from health posts to health centers, and from health centers to district hospitals. Some health centers have a more comprehensive package of services, including a higher number of qualified staff, such as doctors. Health posts primarily offer basic health services, focusing mainly on preventive care and some curative services. They are staffed by health extension workers with limited capacity.

In most communities, transportation options are scarce, so referrals from health posts to health centers usually require walking. Ambulance services are available only for referrals from health centers to hospitals. The Somali region in Ethiopia is considered one of the most disadvantaged regions, with limited infrastructure, including roads, and scattered communities.

We have added an additional sentence in the methods section to clarify this point: “The primary mode of transportation for referrals from HPs to HCs was walking, as transportation options are scarce in most communities. Ambulance services are exclusively available for referrals from health centers to hospitals.”

It would be good to include a workflow diagram for the methodology, so that the approach could be followed in other study areas where detailed geospatial data are lacking

Response: We appreciate the reviewer's valuable insight regarding the applicability of our approach to different contexts and countries. We concur that similar efforts can be successfully transferred by utilizing our methodology. It is worth mentioning that the methodology of accessibility modeling and the approach employed in this study have been extensively described in other literature sources (Hierink et al., 2022), which include openly accessible scripts and dataflows (https://github.com/fleurhierink/Population_Access & <https://zenodo.org/record/7004009>) that can be readily adapted to suit various settings. Additionally, comprehensive documentation is openly available for the AccessMod software (<https://www.accessmod.org/>), further aiding in the understanding and implementation of the methodology. Given the need to maintain a concise methodology section, we believe that referring to the step-by-step approach outlined elsewhere is adequate and will avoid unnecessary lengthiness. We will further ensure that some of the R-scripts used to run the analyses described here are made available upon publication.

Line 68 -- Mention whether travel times were based on walking or car/motorized. This is explained later, but briefly mention here also.

Response: We have added the following two sentences to clarify the scenarios considered in the study:

“In our analyses, we considered both motorized and walking speeds, along with a separate scenario dedicated to walking. However, when addressing referrals, we focused exclusively on the walking scenario. The reasoning behind this choice is elaborated further in the subsequent section.”

Lines 98-100 describe health facility locations in water, etc. Was this due to lack of precision in the location coordinates for some health facilities, or to the coarse spatial resolution of the land use data (grid cells). Explain briefly

Response: Thank you for pointing this out to us. We agree that this can be further clarified. We have added the following sentence to the method section:

“The presence of health facility locations situated on landscape features that pose barriers to the population, such as rivers or lakes, can be attributed to two factors. Firstly, the terrain features used in our analyses, including rivers and roads, are rasterized at an approximate resolution of 100 meters. In certain cases, this resolution may slightly overestimate the width of rivers, leading to the placement of facility coordinates on barriers. Secondly, there may be slight imprecisions or inaccuracies in the GPS coordinates of the facilities. To mitigate the potential impact of these factors on our analyses, we adjust the facility coordinates by relocating them to the nearest land grid cell.”

The Results section is too detailed and relies too much on material in the many supplementary tables and figures. The section is overloaded with statistics that would be appropriate for a governmental report but are not very meaningful for a broad audience of researchers and policymakers. This is especially true in the workforce and workers sections. Tighten up this section by omitting some of the detailed statistics; eliminating many of the supplementary tables and figures; and focusing more on the overall patterns and trends.

Response: We acknowledge the reviewer's feedback regarding the density of information and statistics in the results section. In response to this valuable suggestion, we have made several modifications. Firstly, we have incorporated supplementary figure 3 as the main Figure 2 in the manuscript, thereby eliminating the need for it to be included as supplementary material. Additionally, we have removed supplementary figure 4 and 6 and any references to them in the text.

However, we firmly believe that the remaining supplementary figures in our study provide essential information and highlight the strengths of our research. Specifically, these figures demonstrate the detailed geographical distribution of the health workforce, which can serve as an inspiration for further research and guide policymakers in utilizing similar approaches. These figures showcase the potential of our approach in diverse contexts, particularly considering the direct connection between this field of research and policy and decision-making. Furthermore, these results can contribute to the successful implementation of the OHEP roadmap in other regions of the country.

In discussing referrals (starting on Line 166), you focus on distance, not travel time. Explain – was this done because distance is emphasized in the Roadmap? Also, explain whether distance is Euclidean or based on road/path network. The same paragraph then shifts to travel time, but it is unclear if travel time is based on walking or walking plus motorized.

Response: *We acknowledge the reviewer's concern regarding potential confusion. It is important to clarify that the standard measure employed for implementing the OHEP roadmap is referral distance, not travel time. To address this, we have included an additional sentence in the results section, explicitly stating:*

“Referral were calculated considering both physical distance (in kilometers) and time. The inclusion of the distance metric was essential as it aligns with the standard measure for implementing the OHEP roadmap.”

Lines 238-243. Some of the workforce results are only presented in supplementary tables and only discussed very briefly. For readers, this can be confusing. Either omit some of the supplemental results, or explain briefly in the text. For example, the statement “... nurses make up the biggest group” is vague. Either delete or just give the overall statistic in the text – i.e. Nurses account for X% of the overall HC workforce. Figures 3 and 4 already contain a lot of good information, so just focus on those.

Response: After careful consideration, we acknowledge that the mentioned information and accompanying Supplementary Figure 4 may be redundant and do not provide significant additional depth to the results. As a result, we have decided to remove these lines and the corresponding Figure from the text.

Lines 294-297 are difficult to follow and need to be rewritten. For example, the first sentence refers to the results for two woredas, and it is unclear what the value ranges (e.g. 25-30) refer to. The next sentence

Response: *We agree with the reviewer that the sentence was unnecessarily complex and required improvement. What we intended to convey is that in the Somali region, there are two woredas where over 75% of the population resides more than 2 hours away from either a health post or a health center. Consequently, the presence of Health Extension Workers (HEWs) is crucial to ensure healthcare access to these remote communities. However, it is precisely in these two woredas where there is a shortage of 25-30 and 30-40 HEWs, as indicated by the red arrows on the plot below. We have made revisions in the manuscript to clearly state the following:*

“In two specific woredas, more than 75% of the population resides beyond a 2-hour distance from both health posts and health centers. Consequently, the presence of HEWs is vital to ensure healthcare access for these remote communities through conducting outreach activities. However, it is precisely in these two woredas where there is a shortage of 25-30 and 30-40 HEWs.”

The discussion section also focuses too heavily on detailed findings about specific health service needs in the study area and policy recommendations. Need to cut some of the details and focus more on general findings that may be relevant in similar kinds of places; lessons learned; strengths of the methodology; etc.

Response: We appreciate the reviewer's concern regarding the level of detail provided about the Ethiopian context and the need to emphasize the wider approach. In response, we have added a paragraph to the discussion section that highlights the strengths of our approach in a more general context. This allows us to showcase the broader implications and potential applications of our research beyond the specific Ethiopian setting.

However, we still believe it is essential to situate our study findings within the unique Ethiopian context due to its particularities, such as the OHEP roadmap, pastoralist setting, and health extension worker program. These contextual factors significantly influence the interpretation and practical implications of our research within the Ethiopian healthcare system.

Additional paragraph:

“The main strength of this study lies in its novel approach of coupling geographical accessibility models with facility-level information on health worker staffing. This integration allows for a more realistic assessment of geographical accessibility to specific health worker specialties and provides valuable insights into areas where the health workforce should be strengthened even at the facility level. In contrast, static measures such as the number of health workers per 100,000 population, which are often still the standard, fail to capture the actual geographic coverage of the health workforce. Additionally, this study is among the first of its kind to compare the results against the benchmarks set by the World Health Organization to achieve UHC. The study has shown that even though geographic accessibility of the primary health care system may seem high, the actual availability of staffing to meet the demand in health care may be low. This is especially the case in urban areas where population density is high. This study highlights that solely focusing on geographical access fails to capture the field reality of people seeking care and has highlighted the importance of health extension workers that reach out to remote communities. Unfortunately, our study was limited by the lack of precise data on care utilization, which hindered our ability to quantify the population's specific service requirements and adjust the necessary additional health workforce accordingly. Combining geographic accessibility models with more detailed information on the availability of health services and health workers is key in realistically assessing gaps in service delivery.”

Minor edits:

Lines 42 and 43 – don't need to repeat the abbreviations (HP, HC) here, because they were provided on the previous page

Line 62 – change to: restructuring of the HEP

Lines 126-7 – Briefly explain the UHC Index – percent of what?

Fig. 1 e and f – It's hard to see the gray-tone shading. Might be good to switch to color shading.

Lines 213-15 – “additional” – compared to what? Compared to HCs only? Compared to the situation before HPs?

Lines 234-5 -- potentially to compensate ... unclear – rewrite.

Line 289 – change to: guidelines

Figure 5 – x-axis should be labelled: Number of HEWs (Surplus or Shortage)

Line 322 – optimization of??

Line 329 – change: from the TO of an

Response: *We thank the reviewer for thoroughly checking the manuscript and indicating these minor edits. We have incorporated all changes, which are indicated in blue throughout the manuscript. We have also integrated the suggestions for the figures and have now added green shading for the catchments in Figure 1. The x-axis for Figure 5 has also been updated.*

References

1. UNICEF. *Somali Regional Brief.*

<https://www.unicef.org/ethiopia/media/6521/file/Somali%20regional%20brief.pdf#:~:text=Somali%20region%20is%20one%20of%20the%20government%E2%80%99s%20four,flash%20floods%2C%20locust%20infestations%2C%20diseaseoutbreaks%20and%20intercommunal%20conflict.> (2022).

Reviewers' comments:

Reviewer #1 (Remarks to the Author):

All comments have been addressed.

Reviewer #2 (Remarks to the Author):

Thank you for your thorough and responsive edits

Reviewer #3 (Remarks to the Author):

This is an informative paper that shade light on the feasibility of operationalizing the HEP roadmap in general and in pastoralist settings in particular. I suggest revising the following issues before publication.

1. The Conclusion in the Abstract is mere repetition of the objectives of the paper. Please add take home messages of the paper. It can be like the need for upgrading facilities, strengthening outreach or mobile services, etc.
2. Introduction section, the first paragraph mixes the problem statement as well as the rationale and significance of the study. I would suggest the first paragraph to focus of the problem statement. Accordingly, expand this paragraph to include the national state of inequalities and move the knowledge gap and significance of the study to last paragraph.
3. Methodology, first paragraph, last 3 sentences seem strengths and limitations of the study. Paraphrase it just narrating what you did, combining spatial accessibility and availability of health works. And move the strengths and limitations to Discussion section.
4. The current OHEP program recommends assignment of additional health works at basic health posts. In Somali and Afar regions, health posts are staffed with HEWs and nurses. Why did researchers not include nurses in HR data for HPs in addition to HEWs? Discuss it in the limitations of the study section.
5. Last paragraph of page 11 and first 2 paragraphs of page 12 have contents that should be moved to the Discussion section.
6. The first paragraph of the Discussion section well summarized the key findings. In the subsequent paragraphs, I was expecting researchers to compare each main finding with literature and discuss its implications. Then discuss the strengths and limitations of the study (its implications on the interpretations of findings and efforts to minimize limitations or bias). I feel mix in the flow. For instance, page 19, last 3 sentences seem limitations of the study. In addition, the "Implementation and research challenges" sub-heading seems limitations of the study.

Minor comments

1. page 7, define DEM at its first appearance

2. page 8, last sentence of first paragraph change into past tense
3. Page 11, in the sentence "Only 9% and 16.5% of the HCs cover more than 15,000 people in 1-hour and 2-hours walking travel time respectively (Figure 2)." I think 16.5% is for 1-hour and 9% for 2-hour scenario. Please edit it accordingly.
4. Discussion, paragraph 1, last sentence is vague. Paraphrase it

Reviewer #3 (Remarks to the Author):

This is an informative paper that sheds light on the feasibility of operationalizing the HEP roadmap in general and in pastoralist settings in particular. I suggest revising the following issues before publication.

We highly value the reviewer's dedication in reviewing our manuscript and offering valuable suggestions and comments. We have carefully considered the reviewer's feedback and made the necessary changes throughout the text. We have addressed the important points raised and incorporated the recommended modifications. Below, we have provided responses indicating the specific modifications made in accordance with the reviewer's recommendations.

1. The Conclusion in the Abstract is mere repetition of the objectives of the paper. Please add take home messages of the paper. It can be like the need for upgrading facilities, strengthening outreach or mobile services, etc.

We appreciate the reviewer's feedback regarding the conclusion in our abstract and acknowledge the need to strengthen it. As a result, we have rewritten the conclusion to better address the concerns raised. It now reads as follows:

"Improving accessibility and addressing healthcare worker scarcity are challenges for implementing the primary care roadmap in Ethiopia. Upgrading health posts and centers, providing comprehensive services, and training healthcare workers are crucial. Effective outreach strategies are also needed to bridge the gap and improve accessibility and availability."

2. Introduction section, the first paragraph mixes the problem statement as well as the rationale and significance of the study. I would suggest the first paragraph to focus on the problem statement. Accordingly, expand this paragraph to include the national state of inequalities and move the knowledge gap and significance of the study to last paragraph.

We acknowledge the feedback regarding the introduction and have revised the first paragraph to provide a description of the national health situation in Ethiopia and separate it from the regional problem statement. The introduction now begins with a focus on national health indicators, with the introduction of the Somali region placed towards the end for better coherence and clarity.

The first paragraph now reads:

"Ensuring access to primary healthcare is an essential prerequisite for achieving Universal Health Coverage (UHC), one of the key targets of the United Nations Sustainable Development Goals (SDGs) for 2030^{1,2}. Globally, large gaps and spatial inequalities persist in accessing primary healthcare³⁻⁷, with many communities in sub-Saharan Africa facing considerable distances from healthcare services, especially in sparsely populated rural areas⁸. The burden of inadequate and low-quality healthcare disproportionately affects low- and middle-income countries (LMICs), leading to preventable deaths^{9,10}. Estimates suggest that in 2016, 8.6 million deaths in LMICs could have been avoided if accessible and quality healthcare services were available¹⁰. Being one of the least-developed countries on a global scale and the second most populous country in sub-Saharan Africa, Ethiopia faces notable challenges in healthcare delivery^{11,12}. Despite notable improvements in Ethiopia's health indicators, the country still grapples with challenges such as a heavy burden of communicable and noncommunicable diseases, persistent malnutrition worsened by conflicts and natural disasters, insufficient healthcare coverage and quality, concerns about sustainable healthcare financing, the need for workforce enhancements, gaps in disease surveillance and monitoring, and inadequate preparedness for future health emergencies^{11,12}."

3. Methodology, first paragraph, last 3 sentences seem strengths and limitations of the study. Paraphrase it just narrating what you did, combining spatial accessibility and availability of health works. And move the strengths and limitations to Discussion section.

We removed the sentence from the method section and added a paraphrased version to the discussion. This sentence reads:

"Moreover, while the inclusion of human resources in healthcare accessibility falls short of providing information on dimensions like affordability, acceptability, and accommodation, it does offer a more comprehensive understanding compared to solely considering geographical access."

4. The current OHEP program recommends assignment of additional health works at basic health posts. In Somali and Afar regions, health posts are staffed with HEWs and nurses. Why did researchers not include nurses in HR data for HPs in addition to HEWs? Discuss it in the limitations of the study section.

Unfortunately, this data was not available at the health post level for Somali region. We made a statement in the methods section about this: "The data were originally collected in 2020 and included information on various health worker groups at HCs, and only the number of HEWs at HPs."

To further emphasize this lack of information as a limitation of the study we have added a sentence to the discussion, reading: "Furthermore, our human resource data for HPs was limited and focused solely on the number of HEWs, failing to provide a comprehensive reflection of other crucial health worker categories involved in service delivery at this level, such as nurses."

5. Last paragraph of page 11 and first 2 paragraphs of page 12 have contents that should be moved to the Discussion section.

We acknowledge the reviewer's suggestion to include these parts of the results in the discussion section, as they would be relevant there as well. However, we believe that incorporating them into the discussion might disrupt the flow in the results section and potentially result in unnecessary lengthening of the discussion. By keeping these parts in the results section, we maintain conciseness and clarity. Therefore, we have made the decision to retain them in the results section.

6. The first paragraph of the Discussion section well summarized the key findings. In the subsequent paragraphs, I was expecting researchers to compare each main finding with literature and discuss its implications. Then discuss the strengths and limitations of the study (its implications on the interpretations of findings and efforts to minimize limitations or bias). I feel mix in the flow. For instance, page 19, last 3 sentences seem limitations of the study. In addition, the "Implementation and research challenges" sub-heading seems limitations of the study.

We acknowledge that certain parts of our discussion may not adhere strictly to the conventional structure. To address this concern raised by the reviewer, we have made adjustments to the sub-heading related to the limitations of the study, revising it to "Study Limitations & Implementation Challenges." Additionally, we have included additional limitations of the study as outlined in our previous responses.

Given that our study is among the first to examine geographic accessibility in combination with human resource availability in the Somali region, it is challenging to directly compare all of our findings to previous research. However, we have added a section to compare our results to a recent study conducted by Hendrix et al. (2023), which we have included below for reference.

We believe it is necessary to maintain the rest of the discussion in its current form, as it establishes the context in which our study was conducted and outlines its potential implementation. We hope that these revisions adequately address the concerns raised by the reviewer.

Additional section in discussion:

"A recent study conducted by Hendrix et al. (2023)⁹ also examined the conjunction of geographical accessibility with human resource availability in Ethiopia. However, in this study, the required human resource availability for each facility type was considered to mirror the actual availability of healthcare workers. As demonstrated in this research, it is evident that the actual availability is frequently much lower. However, the results of Hendrix et al. (2023) indeed also show high travel times and low access to healthcare for most of Somali region⁹."

Minor comments

1. page 7, define DEM at its first appearance
2. page 8, last sentence of first paragraph change into past tense
3. Page 11, in the sentence "Only 9% and 16.5% of the HCs cover more than 15,000 people in 1-hour and 2-hours walking travel time respectively (Figure 2)." I think 16.5% is for 1-hour and 9% for 2-hour scenario. Please edit it accordingly.
4. Discussion, paragraph 1, last sentence is vague. Paraphrase it

We thank the reviewer for thoroughly checking and reading the manuscript. Point 1, 2, and 4 have been incorporated into the manuscript. We have also carefully rechecked comment #3 and can confirm that the order of the percentages is correct. It is correct that fewer people will be covered within a 1-hour travel time compared to a 2-hour travel time.